# Demography and productivity during the recovery time sequence of a wild edible bamboo after large-scale anthropogenic disturbance

Noboru Katayama[¤a]*, Osamu Kishida[¤b], Chikako Miyoshi, Shintaro Hayakashi, Kinya Ito, Rei Sakai, Aiko Naniwa, Hiroyuki Takahashi, Kentaro Takagi

Teshio Experimental Forest, Field Science Center for Northern Biosphere, Hokkaido University, Toikanbetsu, Horonobe, Hokkaido, Japan

¤a Current address: General Education, Faculty of Commerce, Otaru University of Commerce, Midori, Otaru, Hokkaido, Japan
¤b Current address: Tomakomai Experimental Forest, Field Science Center for Northern Biosphere, Hokkaido University, Takaoka, Tomakomai, Hokkaido, Japan
* n-kata@res.otaru-uc.ac.jp

**Data Availability Statement:** All relevant data are within the manuscript and its Supporting Information files.

## Abstract

Anthropogenic disturbances in forest management practices can affect wild edible plants. Soil scarification is a large-scale disturbance that may cause long-term reduction in productivity of edible dwarf bamboo, *Sasa kurilensis*, in northern Japan. For their effective and sustainable use, we need to understand the recovery process after such disturbances. At 14 study sites in the Teshio Experimental Forest of Hokkaido University where soil scarification had been conducted between 2 and 44 years prior, the number and stem diameter of old and young (newly emerged, edible) culms was recorded. At sites that were within 11 years of soil scarification, the proportion of old culms (<11%) was lower than in the control area where soil scarification had never been conducted. At sites where more than 15 years had passed since soil scarification, the relative number of old culms was nearly equal to that in control area. Additionally, the number of young culms increased with an increasing number of old culms. These results suggest that recovery of productivity (in term of number) of edible culms may take a few decades. In contrast, the culm diameter of young culms increased linearly with time since soil scarification, but the 95% confidence interval in this relationship suggests that dwarf bamboo can produce thick edible culms soon after soil scarification. These findings will provide useful insights into how to obtain high quality bamboo culms following anthropogenic disturbances in future.

## Introduction

Humans obtain a variety of ecosystem services from ecological environments, e.g., supply and regulation of water, retention and accumulation of soil, and carbon and other elemental storage [1]. Ecosystem services of stably maintained ecological environments can be sustainably

**Funding:** This work was supported by KAKENHI Grant Number 26740045 (NK) and 18K11724 (NK) from Japan Society for the Promotion of Science. The funder had no role in study design, data collection and analysis, decision to publish, or preparation of the manuscript.

**Competing interests:** The authors have declared that no competing interests exist.

utilized; however, natural and anthropogenic disturbances to ecological environments alter the stability of these environments. Intense impact disturbances that exceed the ecological resilience sometimes cause critical damage to ecosystem functioning, which leads to a dramatic decline in ecosystem services [2, 3]. In particular, large-scale human-induced impacts to forests have intensified worldwide in recent decades, and there is increasing concern about the influence of anthropogenic disturbances on forest ecosystems [4, 5]. As humans have come to depend on forest resources through the ages, understanding the impacts of anthropogenic disturbances on ecosystem services in forests is a critical issue.

Provisioning services are one of the major fundamental services in forest ecosystems [6, 7]. Forests provide not only timber products, but also non-timber forest products (NTFPs) such as ecological medicines (i.e., wild medical herbs) and foodstuff (i.e., wild edible plants). NTFPs have been primary medicines and essential foods for humans since ancient times and remain as an important source of income in many nations [8, 9]. While NTFPs principally grow without management by humans, their productivity is greatly influenced by human activities [10, 11] and also depends on environmental conditions [12, 13].

A number of NTFPs are known to be disturbance-dependent species, which are favored to appear in habitats following natural disturbances [6]. Additionally, NTFPs with high compensatory growth can show increased productivity following harvest by humans [14]. Nevertheless, large-scale anthropogenic disturbances, such as over-utilization and/or forest practices, can drastically decrease the abundance and habitat of NTFPs. For sustainable utilization of NTFP resources, we need to understand how large-scale anthropogenic disturbances influence the demography and productivity of NTFPs over time. In this study, we examine the long-term effects of large-scale anthropogenic disturbances on edible dwarf bamboo, *Sasa kurilensis*, based on demography and productivity of the bamboo.

Dwarf bamboos (genus *Sasa*) are prominent understory species in East Asian forests, including Japan [15–17]. Eating newly emerged bamboo culms is part of Asian food customs, and new culms of *S. kurilensis* (but not *S. senanensis*) are one of the most popular wild edible plants in northern Japan. In Hokkaido, the northernmost island of Japan, *S. kurilensis* and *S. senanensis* are the dominant dwarf bamboo species and cover over 89% of the forest area [18]. *Sasa* quickly expands its range widely over forest areas by extending vegetative culms (ramets) [19]. The culms can reach over 2 m in height, completely covering the understory in some cases, and dwarf bamboos have been targeted for eradication because thick stands of dwarf bamboos strongly inhibit establishment and growth of seedlings of commercial trees [20, 21].

Soil scarification has been conducted in Hokkaido, Japan since the mid-20th century to remove understory vegetation together with dwarf bamboo. Soil scarification is a forestry practice of peeling off the soil surface using civil engineering machinery [22, 23], and in most cases, this process leads to the establishment of secondary forests dominated by birches, *Betula* spp., after exposing mineral soil [22, 24]. While there is a growing body of evidence that soil scarification leads to the successful establishment of secondary forests [22, 23, 25, 26], knowledge about the demographic patterns of dwarf bamboo after soil scarification remains limited. Moreover, little is known about how long it takes to recover productivity of edible culms after soil scarification.

The objective of this study is to demonstrate the recovery process of the demography and productivity of *S. kurilensis* after soil scarification. We carried out a field survey in the Teshio Experimental Forest of Hokkaido University, Japan, where soil scarification had been conducted since the 1970s with a record of when and where it had been conducted. In this survey, we selected 14 study sites (disturbed areas) where soil scarification had been conducted between 2 and 44 years previously, and we measured the number and culm diameter (the stem diameter) of both old and young (newly emerged, edible) bamboo culms as demographic and

productivity indexes. Because environmental factors influence the demography and productivity of bamboo, we cannot directly evaluate the time sequence of the recovery process. Thus, we also investigated the demography and productivity of bamboo in neighboring control areas where scarification had not been conducted. Based on analysis of the paired study areas (i.e., disturbed areas vs. control areas), we estimated the duration of recovery for the bamboo population.

## Materials and methods

### Materials

*S. kurilensis* grows in the northern mountainous regions of Japan. Old culms reach a height of about 1.5 to 3 m, and individual culms (ramet) have a lifespan of approximately 8 years, and the individual dwarf bamboo (genet) may have a lifespan of more than 60 years [27, 28]. In early June in northern Japan, several new culms, which are edible, emerge around the old culms. These are known as 'Hime-takenoko' or 'Nemagari-take' and are commercially traded wild vegetables in Japan.

### Study sites

Study sites were selected in the Teshio Experimental Forest of Hokkaido University (44˚54'–45˚06'N, 141˚56'–142˚10'E; area: 22,517 ha), which is located in the northeast forest zone of Japan where the mean annual temperature is 5˚C (max. 35˚C, min. −35˚C) and the annual precipitation is approximately 1000 mm [29]. While the forest overstory mainly consists of deciduous (e.g., *Betula ermanii*) and conifer trees (e.g., *Picea jezoensis*), the understory includes stands of dwarf bamboos (*S. kurilensis* and *S. senanensis*). The stands of *S. kurilensis* and *S. senanensis* are usually monocultures, and they are not comingled in our study sites (N. Katayama, personal observation). Multiple forest fires and various forestry practices, such as soil scarification, had been conducted during the last and current centuries at these sites.

In this study, we chose 14 study sites (1–14) in the Teshio Experimental Forest based on GPS-linked data documenting where and when soil scarifications (anthropogenic disturbance) had been conducted from 1972 to 2014 (S1 Table and S1A Fig). Field surveys were conducted in 2016, which was 2 to 44 years after soil scarifications. Disturbed areas at these sites were essentially untouched after scarification without forestry practice intervention, harvest of old and young bamboo culms or forest fires. Ecological succession in each disturbed area resulted in establishment of stands of birch trees with *S. kurilensis* (S1B–S1E Fig).

### Field surveys

We investigated demography and changes in productivity of *S. kurilensis*, using the 14 study sites (S1 Table and S1A Fig). In July 2016, we set four survey plots (1 × 1 m²) at 10 m inside the boundary of the disturbed areas at study sites 1–9 where more than 11 years had passed since scarification. The plots were separated by at least 5 m. Because the density of bamboo culms was extremely low at study sites 10–14 where less than 6 years had passed after soil scarification (see Results), we set 10 survey plots (1 × 1 m²) in the disturbed areas at these sites in order to obtain sufficient observations of bamboo culms for statistical analyses. Then, we counted the number of bamboo culms in each plot. In this survey, we classified bamboo culms into 'old' (emerged before the current year) and 'young' (emerged in the current year). The number of old culms was considered to indicate the potential for the bamboos to produce edible culms because the density of old culms positively correlated with the number of edible culms [14] (and see Results). Young culms were regarded as the equivalent of edible culms produced in

the current year. In this survey, we measured the stem diameter of all young culms to estimate the quality (culm diameter) of edible culms as culms with greater diameter are generally classified as being more highly qualified as food items. Further, to examine the relationship between the culm diameters of old and young culms, we also measured the stem diameter of 4 old culms, which were randomly selected in each plot (16 culms per area). If there were fewer than 4 old culms in the plot, the stem diameter was measured for all old culms.

To evaluate the demography and productivity of bamboo culms after minimizing any confounding effects due to environmental conditions, we conducted the same survey in 'control areas' at study sites (14 control areas, in total) located within 100 m from the edge of each disturbed area. Based on the GPS-linked data, we chose areas with 'no records' of anthropogenic disturbance including soil scarification and bamboo-harvesting, set four survey plots in each control area, and measured the number and stem diameter of old and young culms by the same method mentioned above. The "original" conditions of old (mother) bamboos would not be different between treatment groups (i.e., disturbed areas vs. control areas) because the number and diameter of old culms, which strongly reflect differences in environmental conditions (N. Katayama, unpublished data), did not differ between treatment groups at more than 24 years after soil scarification (see Results).

## Statistical analysis

To examine changes in the number and stem diameter of old and young culms after soil scarification, we applied general linear mixed models (GLMMs) with an assumption of normal distribution using treatments (control vs. disturbance) and time after the soil scarification as fixed factors. In these models, site identity was used as a random factor. Because we detected significant interactions between time and treatment (see Results), we individually conducted GLMMs to examine the time sequence process on each independent factor (i.e., the numbers and culm diameter of culms) in each treatment (i.e., control and disturbance).

To examine the recovery process of the bamboo after soil scarification in more detail, we calculated the log response ratio (ln[treatment/control]) using numbers and diameters of both old and young culms in the time sequence. The log response ratio is widely used to compare the effect magnitude in manipulation experiments [30], and in our study, to correct for the effects of study site locations (i.e., unknown environmental factors derived from the site locations) on each independent factor. A log response ratio of <0 is taken as a negative effect relative to the control, and conversely, a log response ratio of >0 indicates a positive effect. To calculate the mean and 95% confidence interval (CI) of the log response ratio of the numbers of old and young culms, separate bootstrap models were used with 1000 resampling iterations at each study site. In these models, the number of culms in the disturbed plots (scarification plots) was used as the treatment value and that in the corresponding control plots was taken as the control value. To avoid 0 in the denominator and numerator, 0.1 was added to each value. In addition, we calculated mean and 95% CI of log response ratios of culm diameter using similar separate bootstrap models. In the bootstrap models, data of individual culms instead of the plot mean data were used because there were no or few culms in some plots. In this survey, we used stem diameter data of a total of 379 old and 229 young culms, but we could not carry out bootstrap models of diameter of young culms in disturbed areas at sites 10 and 14 due to too few young culms. Then, general additive models (GAMs) assuming normal distribution were constructed to demonstrate the non-linear relationships between the log response ratios and time since soil scarification using mean log response ratios and years as dependent and independent values, respectively.

To examine the relationships between the number of old and young culms per plot and between the culm diameter per plot, we conducted GLMMs using the site identity as a random factor.

The GLMMs and GAMs were carried out using JMP ver. 14 (SAS Institute Inc., Cary, NC, USA) and the R 'mgcv' package ver. 1.8.17 [31], respectively.

## Results

While the number and diameter of both old and young culms increased significantly after soil scarification (S2 Table and S2 Fig), there were significant interactions between time and treatment (S2 Table). Based on the results of subsequent GLMMs separately analyzed for each treatment, we found that none of the dependent variables were affected by time in the control plots (S3 Table). However, values for each dependent variable in the disturbed plots were significantly increased with time (S3 Table).

In analyses of log response ratios, the GAM demonstrates that the fit smoothing spline (i.e., non-linear relationship) between the relative (i.e., log response ratio) number of old culms and years after soil scarification is a saturation curve ($F_{2.2,2.7} = 41.07$, $P < 0.0001$; Fig 1A), but the average values of the log response ratios were extremely low at sites within 11 years following soil scarification ($<11\%$, compared to control area). The upper ranges of 95% CI of the log response ratios at these sites were lower than the neutral level (ratio = 0). On the other hand, the log response ratios did not significantly differ from 0 at sites surveyed more than 14 years after soil scarification (Fig 1A).

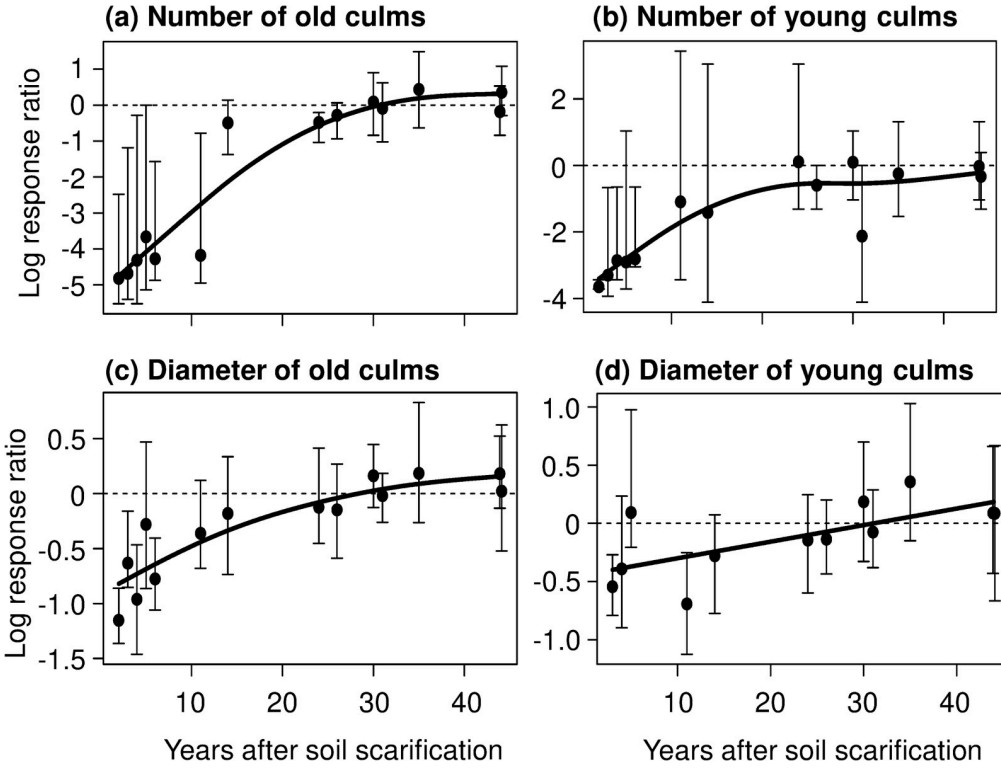

**Fig 1. Log response ratios of number and culm diameter of old and young bamboo culms at 2 to 44 years after soil scarification.** Error bars denote 95% CI from 1000 resampling iterations by bootstrap models, and solid curves indicate non-linear regressions by GAMs. Dotted lines indicate a neutral level of the effects of soil scarification ('log response ratio' = 0). The upper panels show plots of mean data, and the lower panels show individual culm data.

In addition, the relative number of young culms is related to time after soil scarification by a saturation curve (GAM, $F_{2.7,3.3} = 15.29$, $P = 0.0001$; Fig 1B). At sites surveyed within 6 years after soil scarification, the log response ratios were significantly lower than 0 except for at one site (site 11, 5 years after soil scarification), but the log response ratios were not significantly different from 0 at other sites (more than 11 years after soil scarification).

The relationship between the log response ratio of the diameter of old culms and years was also a saturation curve (GAM, $F_{2.0,2.4} = 18.27$, $P < 0.0001$; Fig 1C), and the log response ratios were significantly lower than 0 within 6 years after soil scarification, except at one study site (site 11) and were not different from 0 at sites more than 11 years after soil scarification. On the other hand, the log response ratio of the diameter of young culms gradually increased with time after soil scarification (Fig 1D), and fitting of the smoothing spline (GAM, $F_{1,1} = 8.83$, $P = 0.0136$) was approximately linear. Based on the range of 95% CI of the log response ratios, there was no significant effect of soil scarification on relative culm diameter, except at two sites: study site 9 at 11 years after soil scarification and study site 14 at 2 years after soil scarification.

There was a positive relationship between the number of old and young culms per plot (GLMM, $F_{1,61.7} = 121.02$, $P < 0.0001$; Fig 2A), as well as between the average diameter of old and young culms per plot (GLMM, $F_{1,52.7} = 83.63$, $P < 0.0001$; Fig 2B).

## Discussion

Wild edible plants growing in natural fields are strongly influenced by anthropogenic disturbances, such as human harvesting [14, 32] and forest management practices [33–35]. For instance, removing the edible dwarf bamboo, *S. kurilensis*, during forest management practices (soil scarification) in northern Japan represents a large-scale disturbance that can subsequently decrease its productivity of edible culms for a long time. In this study, we evaluated the long-term impacts of soil scarification on the demography and productivity of *S. kurilensis* through a field survey conducted in a northern Japanese forest.

Our analysis of the log response ratios demonstrated that the recovery process of the number of old culms after soil scarification follows a saturation curve (Fig 1A), suggesting that soil scarification depresses the density of old culms for about 10 years. However, the density of old culms rapidly recovers after some point (by 14 years after soil scarification in our analysis). The density of young culms showed a similar pattern (saturation curve) but the trend was slightly weaker than the response for the density of old culms. To explain this difference in response, we should note that this result is partly a by-product in our statistical analysis; namely, since there was generally a greater number of old culms than young culms in control sites, the log response ratios of old culms tended to be more negative than that for young culms for some period after soil scarification. In any case, the productivity of young (edible) culms was related to the recovery in density of old culms (Fig 1A and 1B and S2A and S2B Fig), which is also indicated by the positive relationship between the number of old and young culms (Fig 2A). Because young bamboo culms emerge from the base of old culms [19, 28], the local density of old culms is a useful index of the productivity and recovery of edible culms in the site. In summary, dwarf bamboo can recover productivity (in terms of 'number' of young culms) within a few decades after soil scarification (Fig 1A and 1B).

A similar trend (saturation curve) was observed in the relative diameter of old culms (Fig 1C). On the other hand, the trend in relative diameter of young culms was a little different; the recovery pattern of the diameter of young culms was an approximately linear relationship (Fig 1D), and there were few sites where soil scarification had a significantly negative impact on the diameter of young culms (except at site 10 and site 14). These results suggest that dwarf

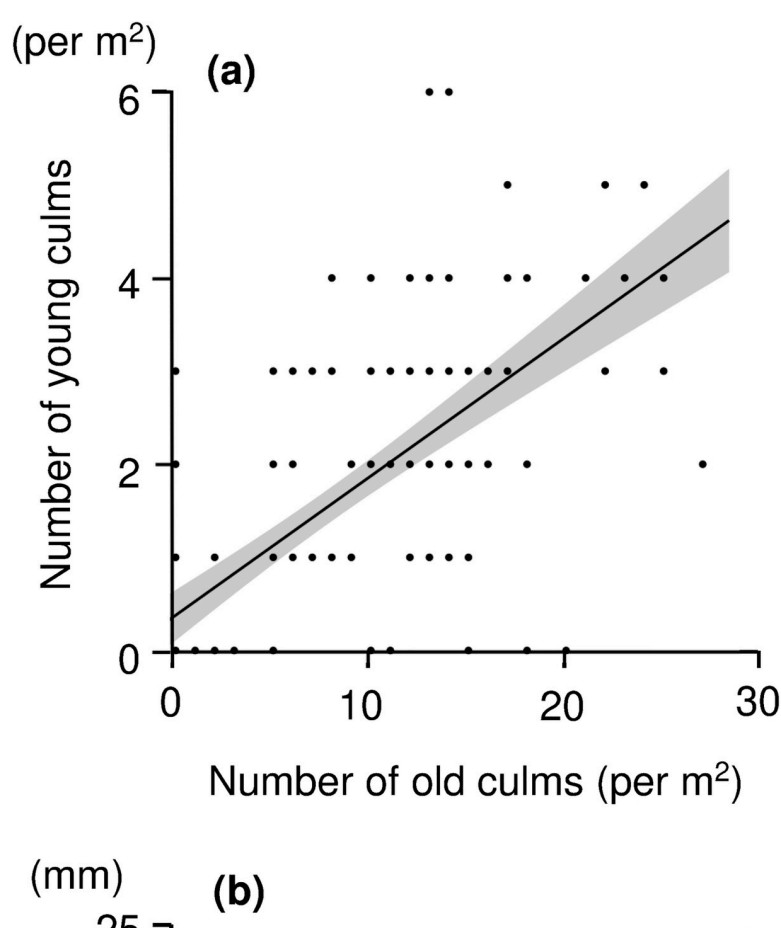

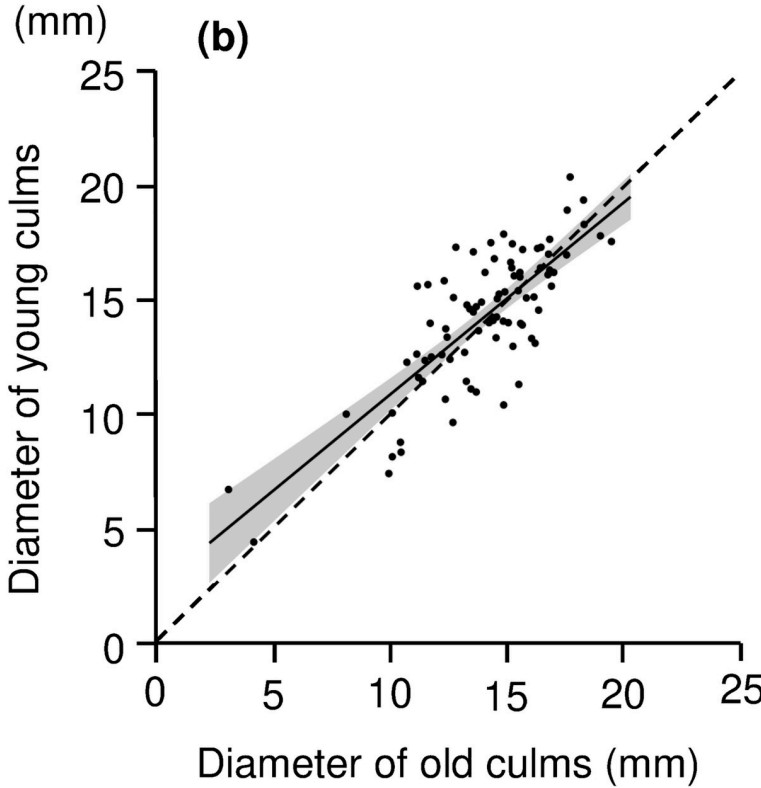

**Fig 2.** Relationships (a) between numbers of young and old culms and (b) between the diameter of young and old culms. Solid lines and grey zones denote linear regression and 95% CI, respectively. The dotted line on the lower figure

has a slope of 1, suggesting no change across years. These figures plot the mean data. Note that several points in the upper figure overlap.

bamboo could produce thick edible culms within a short period of time after soil scarification. Based on the positive relationship between the diameter of old and young culms per plot (Fig 2B), thicker old culms could produce thicker young culms. In addition, in term of 95% CI in this relationship, we can see that thin old culms (especially those with a diameter <10 mm) tended to produce slightly thicker young culms. In this process, bamboo stands with thin old culms would gradually establish stands with thicker young culms. Since the diameter of young culms is an important index of the quality of edible culms, the process of recovering the diameter of bamboo culms after soil scarification is essential for utilizing available ecosystem services from the dwarf bamboo.

While demographic patterns (population size and/or culm diameter) of *S. senanensis* are also likely to change (increase) after soil scarification, populations of *S. senanensis* and *S. kurilensis* are not comingled (i.e., they form monoculture stands) in our study sites. For this reason, we think that the demography of *S. senanensis* may have little effect on that of *S. kurilensis*, and vice versa.

Despite their importance, ecosystem services are rapidly degraded by multiple disturbances [36, 37]. In particular, large-scale anthropogenic disturbances can cause an unexpected change in the status and functioning of ecosystems and have a significant impact on ecosystem services over time [4]. Therefore, we need to understand the exact mechanisms by which disturbances influence the spatial and temporal variation of ecosystem services. In this study, we examined the demography and productivity during the recovery time sequence of an edible bamboo after soil scarification, which is an anthropogenic disturbance, and we demonstrated that recovery of the population size (density) of the dwarf bamboo culms lagged behind in the recovery of culm diameter, indicating that productivity of valuable edible culms may be low for some time after soil scarification.

The abundance of *S. kurilensis* is not declining at present. However, it is uncertain whether this abundance can be maintained in the future. If we look at NTFPs, sudden decline and extinction can occur after large-scale anthropogenic disturbances. Indeed, many NTFPs that were once dominant species have now become endangered due to anthropogenic disturbances (e.g., [7, 38, 39]). For better management, we should examine the potential impacts of anthropogenic disturbances on as many NTFPs (including *S. kurilensis*) as possible. Therefore, even at present, it is worthwhile to examine how *S. kurilensis* recovers following soil scarifications. In addition, our results may be considered as one model case for examining the effect of large-scale anthropogenic disturbances on the productivity of NTFPs that have similar properties (e.g., clonal reproduction via rhizomes) to *S. kurilensis*. The findings of this study can also serve as comparative data for NTFPs with properties that are different from *S. kurilensis*. The findings in this study not only expand our fundamental knowledge of *S. kurilensis* growth but may also make an important contribution toward future forest management efforts, including considerations for management of NTFPs.

## Supporting information

**S1 Fig. Map and photos of disturbed areas in each study site in the Teshio Experimental Forest of Hokkaido University.** (a) Disturbed areas in each study site in the Teshio Experimental Forest of Hokkaido University. Solid zones and lines in the above map indicate site locations and woodland paths, respectively. Numbers on the above map indicate the site ID numbers. (b-e) Photos of disturbed areas in (b) study site 1, (c) study site 5, (d) study site 8 and

(e) study site 11. Photos were taken in May 2014, which is 2 years before the main survey.
(DOCX)

**S2 Fig. Changes in the number and diameter of old and young bamboo culms after soil scarification.** Blue and red circles indicate the mean value per plot in control and scarification treatments, respectively. Dotted blue lines ($P > 0.05$) and solid red lines ($P < 0.05$) denote linear regression in control and scarification treatments, respectively.
(DOCX)

**S1 Table. Number of survey plots at each site.**
(DOCX)

**S2 Table. Statistical results of general linear mixed models (GLMMs) about the effects of soil scarification on old and young culms.**
(DOCX)

**S3 Table. Statistical results of subsequent general linear mixed models (GLMMs) about the changes in status of old and young culms.**
(DOCX)

**S1 File. Dataset in this study.**
(XLSX)

## Acknowledgments

We especially thank Dr. Koya Hashimoto for his helpful comments on our statistical analyses. We are also grateful to the staff at the Teshio Experimental Forest of Hokkaido University for their support during this study.

## Author Contributions

**Conceptualization:** Noboru Katayama, Osamu Kishida, Kentaro Takagi.

**Data curation:** Noboru Katayama.

**Formal analysis:** Noboru Katayama.

**Funding acquisition:** Noboru Katayama.

**Investigation:** Noboru Katayama, Chikako Miyoshi, Shintaro Hayakashi, Kinya Ito, Rei Sakai, Aiko Naniwa, Hiroyuki Takahashi.

**Methodology:** Noboru Katayama, Osamu Kishida, Kentaro Takagi.

**Project administration:** Noboru Katayama.

**Writing – original draft:** Noboru Katayama.

**Writing – review & editing:** Noboru Katayama.

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
