## [Decision Letter · Decision Letter 0]

21 Sep 2020

PONE-D-20-20261

Demography and productivity during the recovery time sequence of a wild edible bamboo after large-scale anthropogenic disturbance

PLOS ONE

Dear Dr. Katayama,

Thank you for submitting your manuscript to PLOS ONE. After careful consideration, we feel that it has merit but does not fully meet PLOS ONE’s publication criteria as it currently stands. Therefore, we invite you to submit a revised version of the manuscript that addresses the points raised during the review process.

We look forward to receiving your revised manuscript.

Kind regards,

Xiao Guo, Ph.D.

Academic Editor

PLOS ONE

Journal Requirements:

Reviewers' comments:

Reviewer's Responses to Questions

**Comments to the Author**

1. Is the manuscript technically sound, and do the data support the conclusions?

Reviewer #1: No

Reviewer #2: Yes

2. Has the statistical analysis been performed appropriately and rigorously? 

Reviewer #1: Yes

Reviewer #2: Yes

3. Have the authors made all data underlying the findings in their manuscript fully available?

Reviewer #1: Yes

Reviewer #2: Yes

4. Is the manuscript presented in an intelligible fashion and written in standard English?

Reviewer #1: Yes

Reviewer #2: Yes

5. Review Comments to the Author

Reviewer #1: The growth of bamboo forest is affected by soil condition, mother bamboo condition and bamboo shoot harvest.

In this study, only the soil condition was discussed, and the other two aspects were ignored. The control and treatment groups were quite different in the condition of mother bamboo and bamboo shoot harvest.

As a bamboo shoot forest, its growth status is not only related to soil, but also closely related to human bamboo shoot harvesting behavior.

Generally, bamboo shoot harvesting and soil scarification occur simultaneously in the bamboo forest managed by shoot use forest.

In this paper, the author only considered the soil scarification factors, and did not consider the bamboo shoot harvesting. Bamboo shoot harvesting can affect the growth of bamboo forest more than soil scarification .

This kind of artificially managed bamboo forest is often accompanied by the felling of old bamboo, which affects the quantity of old bamboo and new bamboo, and also leads to the difference of DBH between new bamboo and old bamboo.

In this paper, the number of old bamboos in the treatment group was significantly different from that in the control group.

As a bamboo species with 8-year-old individual bamboo plant, it is of little significance to study the soil scarification for more than 40 years. Unless it is proved that the physical and chemical properties of soil have changed with the increase of soil scarification time, else, the significance of 20 years and 40 years is not significant. This difference is largely caused by the management behavior of harvesting bamboo shoots and cutting down old bamboo.

The sample plot area of bamboo survey is 1 * 1m, which is too small. In China, the investigated area of Moso bamboo forest is not less than 400 square meters, and the number of small bamboo species is not less than 2 * 2 meters.

Generally speaking, the conclusion of this paper is unreasonable.

Reviewer #2: The manuscript is well organized and written. Soil scarification is one important measure for the secondary forest management. However, soil scarification has been conducted to remove dwarf bamboos rather to cultivate them. Therefore, the dwarf bamboos recovering from the field after soil scarification might be influenced greatly by the environment around. Accordingly, the description of experimental site should be more clear to avoid such a confusion. Moreover, the area of survey plot as 1*1 m2 is suitable to demonstrate the demograhpy and productivity of bamboos?

6. PLOS authors have the option to publish the peer review history of their article (what does this mean?). If published, this will include your full peer review and any attached files.

Reviewer #1: No

Reviewer #2: No

---

## [Author Response · Author response to Decision Letter 0]

1 Oct 2020

[Reviewer #1]

[Comment]

The growth of bamboo forest is affected by soil condition, mother bamboo condition and bamboo shoot harvest. In this study, only the soil condition was discussed, and the other two aspects were ignored. The control and treatment groups were quite different in the condition of mother bamboo and bamboo shoot harvest. 

[Response]

Thank you very much for this comment. We understand that these concerns arise due to the lack of explanation of the study sites. Our study was carried out in an experimental forest for research on forestry, not a commercial forest for the harvest bamboos. Therefore, the traffic of people to harvest bamboos was fundamentally restricted. In addition, we carefully selected undisturbed study sites after soil scarification (please see lines 139-140). There was no record of the harvest of young and old bamboo culms in these sites and, therefore, we could ignore the effects of harvesting on the present results. Another concern about the condition of mother bamboos is likely your misunderstanding. The "original" condition of mother bamboos would not differ between treatment groups (i.e., disturbed areas vs. control areas) because "each" control area was located within the study site (100 m from the edge of "each" disturbed area). The paired areas were selected in this manner in order to avoid any confounding effects of environmental conditions (including the state of the mother bamboo). In fact, there were no significant differences in the numbers and stem diameters of old as well as young culms between treatment groups in sites where more than 24 years had passed after soil scarification (please see Figure 1). This indicates that the bamboo stands had recovered by the time of the study (24 years after soil scarification), and the "original" status of mother bamboos was similar between disturbed and control areas in our study. We added this statement in lines 165-176.

[Comment]

As a bamboo shoot forest, its growth status is not only related to soil, but also closely related to human bamboo shoot harvesting behavior.

Generally, bamboo shoot harvesting and soil scarification occur simultaneously in the bamboo forest managed by shoot use forest.

In this paper, the author only considered the soil scarification factors, and did not consider the bamboo shoot harvesting. Bamboo shoot harvesting can affect the growth of bamboo forest more than soil scarification.

[Response]

Please see the response above describing the history and use conditions of the forest of this study site as well as the selection of sampling areas within the forest.

[Comment]

This kind of artificially managed bamboo forest is often accompanied by the felling of old bamboo, which affects the quantity of old bamboo and new bamboo, and also leads to the difference of DBH between new bamboo and old bamboo.

[Response]

First, please understand that our study sites were not set in an artificially managed bamboo forest but in an experimental forest where bamboo succession occurred naturally. As suggested by your comment, litter of old culms would affect the productivity and stem diameter (in similar sense to DBH) of bamboo culms via nutrient cycling. The amount of litter just after soil scarification would be small because soil scarification removes the above- and below-ground tissues of bamboos. Of course, the recovery process includes changes in litter amount, but the main focus in this study is not to evaluate the effects of "individual" processes on the dynamics of bamboos but to demonstrate "overall" demography and productivity after soil scarification. We consider that Figure 1 shows these results, which support this study approach.

[Comment]

In this paper, the number of old bamboos in the treatment group was significantly different from that in the control group.

[Response]

Until 24 years had passed after soil scarification, there were significant differences in the number of old culms between treatment groups (Figure 1) because the bamboo stands were indeed in the recovering process. After that time, there were no significant differences in the number of old culms between treatment groups, indicating that the bamboo stands had recovered completely. Thus, differences are meaningful and not the result of poor study design.

[Comment]

As a bamboo species with 8-year-old individual bamboo plant, it is of little significance to study the soil scarification for more than 40 years. Unless it is proved that the physical and chemical properties of soil have changed with the increase of soil scarification time, else, the significance of 20 years and 40 years is not significant. This difference is largely caused by the management behavior of harvesting bamboo shoots and cutting down old bamboo.

[Response]

We realize that the explanation of life history of the dwarf bamboo (Sasa kurilensis) was insufficient here. The lifespan of an individual "culm" (each ramet) was actually about 8 years, but the individual dwarf bamboo (genet) was longer lived and may have a lifespan of more than 60 years. We added this statement in line 119. In addition, please check again to confirm that we show "no differences" in the number of old and young culms in the sites where more than 24 years had passed after soil scarification (please see Figure 1).

[Comment]

The sample plot area of bamboo survey is 1 * 1m, which is too small. In China, the investigated area of Moso bamboo forest is not less than 400 square meters, and the number of small bamboo species is not less than 2 * 2 meters.

[Response]

Thank you for your comment. Because the Mosso bamboo has a large body (culm) size, a larger sized plot may be necessary to survey the demography. However, our target bamboo was a dwarf bamboo that forms condensed colonies. The number of old culms was frequently over 10 per m2 (average density of old culms in control area was 13.4 culms/m2; please see S2 Fig). As you suggest, the plot size may be a bit small, but even so, we could demonstrate a meaningful recovery process (Figure 1). In addition, please confirm that we prepared at least 4 plots per site. Thus, the demography of the target bamboo was checked in at least a 2×2 meter square area at each site.

[Comment]

Generally speaking, the conclusion of this paper is unreasonable.

[Response]

We appreciate your comment, but we do not understand how to better express this conclusion based on your comment. Thus, we removed the last paragraph (conclusion) from the manuscript.

[Reviewer #2]

[Comment]

The manuscript is well organized and written. Soil scarification is one important measure for the secondary forest management. However, soil scarification has been conducted to remove dwarf bamboos rather to cultivate them. Therefore, the dwarf bamboos recovering from the field after soil scarification might be influenced greatly by the environment around. Accordingly, the description of experimental site should be more clear to avoid such a confusion. 

[Response]

Thank you very much for your comment. We apologize for the inadequate explanation of our study sites. In this study, each control area was set adjacent to each corresponding disturbed area in order to ensure similar environmental conditions between the paired areas. Actually, the number and diameter of old culms, which strongly reflect the differences in environmental conditions (N. Katayama, in review), did not differ between treatment groups at more than 24 years after the soil scarification. We added this statement in lines 165-176.

[Comment]

Moreover, the area of survey plot as 1*1 m2 is suitable to demonstrate the demograhpy and productivity of bamboos?

[Response]

Thank you very much for your comment. Please see our response to a similar comment from Reviewer #1.

We hope that our modifications adequately address all of the comments of the reviewers. We hope that the revised manuscript is now acceptable for publication in PLoS ONE.

---

## [Decision Letter · Decision Letter 1]

30 Oct 2020

PONE-D-20-20261R1

Demography and productivity during the recovery time sequence of a wild edible bamboo after large-scale anthropogenic disturbance

PLOS ONE

Dear Dr. Katayama,

Thank you for submitting your manuscript to PLOS ONE. After careful consideration, we feel that it has merit but does not fully meet PLOS ONE’s publication criteria as it currently stands. Therefore, we invite you to submit a revised version of the manuscript that addresses the points raised during the review process.

We look forward to receiving your revised manuscript.

Kind regards,

Xiao Guo, Ph.D.

Academic Editor

PLOS ONE

Additional Editor Comments (if provided):

The authors have revised the manuscript according to the comments of two reviewers. The manuscript has been greatly improved. However one reviewer still has a concern about the English language of this manuscript. I agree with the reviewer and therefore recommend the manuscript be formally accepted after language polishment. A certificate of English editing by a professional company will be greatly appreciated. 

Reviewers' comments:

Reviewer's Responses to Questions

**Comments to the Author**

1. If the authors have adequately addressed your comments raised in a previous round of review and you feel that this manuscript is now acceptable for publication, you may indicate that here to bypass the “Comments to the Author” section, enter your conflict of interest statement in the “Confidential to Editor” section, and submit your "Accept" recommendation.

Reviewer #1: All comments have been addressed

Reviewer #2: All comments have been addressed

2. Is the manuscript technically sound, and do the data support the conclusions?

Reviewer #1: Yes

Reviewer #2: Yes

3. Has the statistical analysis been performed appropriately and rigorously? 

Reviewer #1: Yes

Reviewer #2: Yes

4. Have the authors made all data underlying the findings in their manuscript fully available?

Reviewer #1: Yes

Reviewer #2: Yes

5. Is the manuscript presented in an intelligible fashion and written in standard English?

Reviewer #1: Yes

Reviewer #2: Yes

6. Review Comments to the Author

Reviewer #1: (No Response)

Reviewer #2: In this version, all comments of the original were addressed soundly. Therefore, the manuscript is recommended to be published with a language polishment.

7. PLOS authors have the option to publish the peer review history of their article (what does this mean?). If published, this will include your full peer review and any attached files.

Reviewer #1: No

Reviewer #2: No

---

## [Author Response · Author response to Decision Letter 1]

10 Nov 2020

[Editor]

The authors have revised the manuscript according to the comments of two reviewers. The manuscript has been greatly improved. However one reviewer still has a concern about the English language of this manuscript. I agree with the reviewer and therefore recommend the manuscript be formally accepted after language polishment. A certificate of English editing by a professional company will be greatly appreciated. 

[Reviewer #2]

[Comment]

In this version, all comments of the original were addressed soundly. Therefore, the manuscript is recommended to be published with a language polishment.

[Response]

We are very glad to hear these comments, and we apologize for the poor English. The English in this manuscript was carefully checked again by a science editor before the resubmission of this revision. I hope that the revised manuscript is now acceptable for publication in PLOS ONE.

Thank you again in advance for your kind consideration of this manuscript.

Noboru Katayama

---

## [Editor Report · Decision Letter 2]

16 Nov 2020

Demography and productivity during the recovery time sequence of a wild edible bamboo after large-scale anthropogenic disturbance

PONE-D-20-20261R2

Dear Dr. Katayama,

We’re pleased to inform you that your manuscript has been judged scientifically suitable for publication and will be formally accepted for publication once it meets all outstanding technical requirements.

Kind regards,

Xiao Guo, Ph.D.

Academic Editor

PLOS ONE

Additional Editor Comments (optional):

The English language in this manuscript has been carefully checked and edited by a science editor. I recommend the manuscript be accepted for publication in its current form in PLoS ONE.
---

## [Editor Report · Acceptance letter]

18 Nov 2020

PONE-D-20-20261R2 

Demography and productivity during the recovery time sequence of a wild edible bamboo after large-scale anthropogenic disturbance 

Dear Dr. Katayama:

I'm pleased to inform you that your manuscript has been deemed suitable for publication in PLOS ONE. Congratulations! Your manuscript is now with our production department. 

Kind regards, 

on behalf of

Dr. Xiao Guo 

Academic Editor

PLOS ONE